# Diagnostic Performance of ACR and Kwak TI-RADS for Benign and Malignant Thyroid Nodules: An Update Systematic Review and Meta-Analysis

**DOI:** 10.3390/cancers14235961

**Published:** 2022-12-02

**Authors:** Yun Jin Kang, Gulnaz Stybayeya, Ju Eun Lee, Se Hwan Hwang

**Affiliations:** 1Department of Otolaryngology-Head and Neck Surgery, Yeouido St. Mary’s Hospital, College of Medicine, The Catholic University of Korea, Seoul 07345, Republic of Korea; 2Department of Physiology and Biomedical Engineering, Mayo Clinic, Rochester, MN 55902, USA; 3Department of Otolaryngology-Head and Neck Surgery, Bucheon Saint Mary’s Hospital, College of Medicine, The Catholic University of Korea, Seoul 14647, Republic of Korea

**Keywords:** thyroid cancer, ultrasonography, fine-needle aspiration biopsy, meta-analysis, thyroid nodules, systematic review, diagnostic imaging

## Abstract

**Simple Summary:**

This meta-analysis determined the optimal cut-off value for differentiating benign and malignant thyroid nodules in two risk stratification systems (ACR and Kwak TI-RADS) and compared their diagnostic performance. Both systems showed good diagnostic performance. TR4 and 4B were estimated as optimal cut-off values for ACR and Kwak TI-RADS, respectively, but the cut-off values can be adjusted in consideration of changes in sensitivity and specificity.

**Abstract:**

(1) Background: To determine the optimal cut-off values of two risk stratification systems to discriminate malignant thyroid nodules and to compare the diagnostic performance; (2) Methods: True and false positive and negative data were collected, and methodological quality was assessed for forty-six studies involving 39,085 patients; (3) Results: The highest area under the receiver operating characteristic (ROC) curve (AUC) of ACR and Kwak TI-RADS were 0.875 and 0.884. Based on the optimal sensitivity and specificity, the highest accuracy values of ROC curves or diagnostic odds ratios (DOR) were taken as the cut-off values for TR4 (moderate suspicious) and 4B. The sensitivity, specificity, DOR, and AUC by ACR (TR4) and Kwak TI-RADS (4B) for malignancy risk stratification of thyroid nodules were 94.3% and 96.4%; 52.2% and 53.7%; 17.5185 and 31.8051; 0.786 and 0.884, respectively. There were no significant differences in diagnostic accuracy in any of the direction comparisons of the two systems; (4) Conclusions: ACR and Kwak TI-RADS had good diagnostic performances (AUCs > 85%). Although we determined the best cut-off values in individual risk stratification systems based on statistical assessment, clinicians can adjust the optimal cut-off value according to the clinical purpose of the ultrasonography because raising or lowering cut-points leads to reciprocal changes in sensitivity and specificity.

## 1. Introduction

Thyroid nodules are relatively common in the general population, and about 10% of thyroid nodules have a risk of malignancy with increasing prevalence [1,2]. Since malignant and benign thyroid nodules differ in treatment and prognosis, early differentiation between benign and malignant thyroid nodules is important [3].

Currently, ultrasound stratification of thyroid nodules is a fast, primary, and easy-to-use diagnostic tool for thyroid nodule; it is also non-invasive and inexpensive [3]. The characteristics of benign and malignant nodules on ultrasound are different, which can be interpreted and misunderstood differently depending on the experience of the examiner or the image obtained [4]. In most ultrasound examinations, fine needle aspiration (FNA) is recommended to differentiate malignancies [5,6]. However, excessive FNA is not recommended [6] if the thyroid nodule is too small or too large, because it may be difficult to diagnose [6]. Therefore, it is important to confirm the recommended thyroid nodule size for FNA, and comparative studies on its diagnostic value are needed.

The thyroid imaging reporting and data system (TI-RADS) is used to objectively classify thyroid nodules and recommend treatment [7]. The five classification systems currently used most often are: (1) TI-RADS from the American College of Radiology (ACR TI-RADS), (2) Kwak (Kwak TI-RADS), (3) the Korean Thyroid Association or the Korean Society of Thyroid Radiology guidelines, (4) the European Thyroid Association guidelines, and (5) the American Thyroid Association guidelines [3,4,8,9,10,11,12]. Although these guidelines are effective for diagnosing thyroid nodules, it is not clear which is the best diagnostic tool [13]. Furthermore, to the best of our knowledge, there are no meta-analyses comparing two stratification systems for diagnostic efficiency including cut-off values.

In this study, we compared the diagnostic efficiency of ACR TI-RADS and Kwak TI-RADS for diagnosing thyroid nodules and we estimated the optimal cut-off values for each ultrasound stratification system.

## 2. Materials and Methods

### 2.1. Study Protocol and Registration

This systematic review and meta-analysis was conducted in accordance with the Preferred Reporting Items for Systematic Reviews and Meta-Analyses (PRISMA) guidelines; it also followed the recommendations provided for optimal literature search for systematic reviews in surgery. The study protocol was prospectively registered on the Open Science Framework (https://osf.io/fykwr/ (accessed on 2 October 2022)).

### 2.2. Literature Search Strategy

All clinical studies retrieved up to August 2022 from the databases of PubMed, SCOPUS, Embase, Web of Science, and the Cochrane Central Register of Controlled Trials were analyzed. The search terms were: thyroid, thyroid nodule, thyroid neoplasm, malignancy, thyroid cancer, diagnostic imaging, diagnostic performance, diagnostic value, ultrasonography, diagnosis, ultrasound, diagnostic value, ultrasonography, ultrasound risk stratification system, imaging, reporting systems, thyroid imaging reporting and data system, TI-RADS, TIRADS, indeterminate, American College of Radiology guideline, ACR TI-RADS, European Thyroid Association guideline, EU TI-RADS, American Thyroid Association (ATA) guidelines, thyroid, ultrasound classifications, Korean Thyroid Association guideline (K TI-RADS), and Kwak TI-RADS. The references of the studies extracted through the search were also considered to ensure that there were no missing studies except duplicates. Two authors (YJK and SHH) evaluated the title, abstract and, if necessary, the full manuscript. Studies not related to risk stratification system for thyroid nodule using ultrasonography were excluded.

### 2.3. Selection Criteria

The inclusion criteria were studies with: patients undergoing ultrasound studies of thyroid nodules, prospective or retrospective studies, comparison of ultrasound studies with cytologic or histologic findings, and data from sensitivity and specificity analyses. Exclusion criteria were: review articles, case reports, studies of other neck diseases (e.g., lymphadenitis or neck mass), articles not written in English, and articles without adequate data to determine the diagnostic value of imaging studies.

### 2.4. Data Extraction and Risk of Bias Assessment

The articles included in this study were subjected to data extraction, wherein data were organized into a standardized form [14]. Analysis results were diagnostic accuracy (diagnostic odds ratio (DOR)), summary receiver operating characteristic (SROC) curve, and area under the curve (AUC). DOR was calculated using the following parameters: true positive, false positive, false negative, and true negative, and it was used to assess diagnostic accuracy with a 95% confidence interval from random-effects models, considering both within- and between-study variation. Higher DOR values (ranging from 0 to infinity) indicated better diagnostic performance. The SROC approach is considered the best method for meta-analysis and generates paired sensitivity and specificity estimates [6,11,15,16,17,18,19,20,21,22,23,24,25,26,27,28,29,30,31,32,33,34,35,36,37,38,39,40,41,42,43,44,45,46,47,48,49,50,51,52,53,54,55]. As the discriminatory power of a test increases, the SROC curve more closely approaches the top left corner of the receiver operating characteristic curve (ROC) space (i.e., the point where sensitivity and specificity both equal 1 (100%)) [56]. AUC is a value between 0 and 1, and the higher AUC value means better diagnostic performance. An AUC of >0.9~1.0 is considered excellent diagnostic accuracy, >0.8~0.9 is good diagnostic accuracy, >0.7~0.8 is fair diagnostic accuracy, >0.6~0.7 is poor diagnostic accuracy, and ≤0.6 is interpreted as a failed diagnosis [57].

We extracted the following data from the included studies: number of patients, correlations of scores measured in endoscopy and CT, true positive values, true negative values, false positive values, and false negative values. The Quality Assessment of Diagnostic Accuracy Studies version 2 tool (QADAS-2) was used to evaluate methodological quality (e.g., risk of bias) [58]. For the definition of true positive and negative, guideline category < cut-off value was regarded as “test negative” and guideline category ≥ cut-off value as “test positive”. Therefore, “benign” lesions classified as <cut-off were regarded as true negatives and “non-benign” lesions classified as ≥cut-off value were regarded as true positives. Accordingly, the sensitivity, specificity, and DOR were calculated with reference to the final results based on pathological examination, FNA cytology, and follow-up. ROC curve analyses and AUC were used to assess the effectiveness of guidelines in differentiating benign from malignant thyroid nodules [59].

### 2.5. Statistical Analyses and Outcome Measurements

We used R statistical software version 3.6.1 (R Foundation for Statistical Computing, Vienna, Austria) for meta-analysis. Q statistic was used for homogeneity analysis to evaluate heterogeneity. TI-RADS categories were proposed by Kwak et al. to classify thyroid nodules as 2 (benign lesions), 3 (no suspicious ultrasound features), 4a (one suspicious ultrasound feature), 4b (two suspicious ultrasound features), 4c (three or four suspicious ultrasound features), or 5 (five suspicious ultrasound features) according to the risk estimates of malignancy [4]. Ultrasound features in the ACR TI-RADS are categorized as benign (TR1, 0 point), not suspicious (TR2, 2 points), mildly suspicious (TR3, 3 points), moderately suspicious (TR4, 4–6 points), or highly suspicious (TR5, ≥7 points) for malignancy [12]. Diagnostics accuracy in individual risk stratification systems (ACR TI-RADS and Kwak TI-RADS) was assessed based on the use of different cut-off values. Forest plots show sensitivity, specificity, and SROC curves.

## 3. Results

### 3.1. Search and Study Selection

Forty-six studies with 39,085 participants were included in the analyses (Figure 1). Study characteristics are shown in Appendix A, and the results for bias assessment are presented in Appendix A.

### 3.2. Diagnostic Accuracy in Various Ultrasound Risk Stratification Systems

Diagnostic efficacy and ROC curves for the two guidelines according to the various cut-off values are shown in Table 1 and Table 2, respectively.

In Kwak TI-RADS categories, sensitivity changed from 14–99% (highest in 4a) and specificity showed the inverse association (99–27%; highest in 5) according to the different cut-off values (categories). ROC analysis and DOR showed that the best diagnostic cut-off values of Kwak TI-RADS had 4b in common. A cut-off in the screening test was chosen to minimize the rate of false negatives rather than reducing false positives because this would be appropriate for conditions in which misdiagnosing and treating someone as sick is better than missing truly sick individuals [60]. Based on the statistical considerations, the best cut-off value of Kwak TI-RADS was category 4b with 96.4% sensitivity and 53.7% specificity. However, in practical considerations, if the sensitivity or specificity of a screening test were considered to be either too high or too low, they could be adjusted by raising or lowering cut-off values [61]. It has also been suggested that a more appropriate sensitivity and specificity value would have been approximately 73% for both, and therefore, incidentally, similar to the values obtained by other researchers [62,63]. Therefore, 4c could be also be a good cut-off value for Kwak TI-RADS because balanced sensitivity and specificity could be more suitable for screening tests.

In ACR TI-RADS categories, sensitivity changed to 71.0–98.8% (highest in TR3), and specificity showed the inverse association (86.9–23.7%; highest in TR5) according to the different cut-off values (categories). ROC analysis and DOR showed that the best diagnostic cut-off values of ACR TI-RADS were TR5 and TR4, respectively. Although a test with high AUC is statistically considered “better” than one with lower AUC, AUC lacks clinical interpretability because it does not reflect the practical gains and losses to individual patients by diagnostic tests [64]. A cut-off in the screening test has been chosen to minimize the rate of false negatives rather than reducing false positives [1]. Accordingly, the best cut-off value for ACR TI-RADS was category TR4 with 94.3% sensitivity and 52.2% specificity. However, TR5 would also be a good cut-off value for ACR TI-RADS because balanced sensitivity and specificity could be more suitable for screening tests.

### 3.3. Direct Comparison of Diagnostic Performance for Predicting Thyroid Malignancy with the Two TIRADS

Only 11 studies that evaluated the diagnostic accuracy of the two guidelines in the same lesions or patients were included for direct comparison. The ROC curves of the cut-off values for Kwak (4b) and ACR TI-RADS (TR4) indicated that there was no significant difference between the two guidelines (Kwak TI-RADS (AUC 0.842) and ACR TI-RADS (AUC 0.846)) in diagnostic performance for thyroid malignancy (Table 3). Additionally, they both had the statistically similar and high diagnostic performances for sensitivity (Kwak 0.9727, ACR 0.9760, *p* value 0.7572), specificity (Kwak 0.5641, ACR 0.5025, *p* value 0.1916), and DOR (Kwak 44.0619, ACR 38.2503, *p* value 0.6896). Another analysis of a different set of cut-off values for Kwak (4c) and ACR TI-RADS (TR5) also showed no significant difference between the two guidelines (Kwak TI-RADS (AUC 0.89) and ACR TI-RADS (AUC 0.892)) (Table 4). Both studies reported statistically similar high diagnostic performance by sensitivity (Kwak 0.7517, ACR 0.7824, *p*-value 0.6537), specificity (Kwak 0.8754, ACR 0.8544, *p*-value 0.4122), and DOR estimates (Kwak 20.9770, ACR 21.3275, *p*-value 0.9597).

## 4. Discussion

Nodule number, size, calcification, and echo pattern from ultrasound images are considered when classifying thyroid nodules according to TI-RADS [3,7] to differentiate benign and malignant thyroid nodule to determine whether FNA is required [3,13]. The ultrasound stratification systems help to avoid unnecessary FNA in cases when the thyroid nodule is too small, too large, or when benign versus malignant status is ambiguous. However, the FNA recommendation threshold is different for each system, and the results of the studies analyzing the diagnostic effect are not consistent [3,4].

Many meta-analyses have compared different ultrasound risk stratification systems for thyroid nodules [3,4,65,66,67,68], but our study analyzed the diagnostic effect in more detail including cut-off values for two risk stratification systems. In addition to identifying the optimal cut-off value, we directly compared two stratification systems at different cut-off values with an AUC > 0.8 for high specificity and sensitivity.

Other meta-analysis results related to Kwak TI-RAD showed high overall sensitivity and low specificity [65,68], which was associated with a high cut-off of 4a and 4b [65]. In our study, as the cut-off value increased, sensitivity decreases and specificity tended to increase. In other words, if the cut-off is high, it is better to rule out an increase in the number of benign thyroid nodule diagnoses and to reduce unnecessary surgeries. However, there was no significant difference in direct comparison between the best cut-offs in Kwak TI-RADS (4b) and ACR TI-RADS (TR4). From a direct comparison of Kwak TI-RADS (4b and 4c) and ACR TI-RADS (TR4 and TR5), all have an AUC greater than 0.8, which is a theoretically effective diagnostic tool [69].

In our study, both Kwak TI-RADS (4b and 4c) and ACR TI-RADS (TR4 and TR5) had high diagnostic performance, with no significant difference between them. However, although both are point-based systems that have high accuracy and may require complex analysis and calculations, they are internally different [70]. Because TI-RADS should be able to reduce the subjective effect of ultrasound and provide a standard for diagnosis and treatment, further research is needed for the best diagnostic performance with the best cut-off value. For FNA recommendation, further studies on the size threshold are needed. ACR TI-RADS has good diagnostic performance at the cut-off value of TR4 and TR5, but the nodule size threshold for FNA was also clinically important in our study. Thresholds of categories 3, 4, and 5 in ACR TI-RADS were 2.5, 1.5, and 1 cm, which was reported as an effective criterion to reduce unnecessary FNA [71].

There are several limitations and factors explaining the high heterogeneity. First, a selection bias for patients may have occurred. Among the included studies, there were studies that included more malignant thyroid nodules than benign. Second, because thyroid nodule diagnosis is ultimately determined by pathologic or cytologic results, bias may occur depending on clinicians or diagnostic tools. Third, the various experience of radiologists as commonly described in included studies can induce high heterogeneity. However, difficult cases may be accompanied by the supervision of more experienced radiologists [72]. Additionally, intermediate categories were not tested in ACR TI-RADS, and low agreement in cytologic pathology may be expected [65]. Fourth, most included studies were retrospective studies. Prospective and multicenter studies need to be included to reduce bias. Lastly, papillary thyroid cancer is related to BRAF V600 mutation. However, correlation with the ultrasound stratification system and mutation was not considered. The possibility of mutation is low in intermediate nodules, but should be considered.

## 5. Conclusions

ACR TI-RADS and Kwak TI-RADS are both effective for differential diagnosis of benign and malignant thyroid nodules with an AUC of 85% or higher. However, since the change in statistically confirmed optimal cut-off value is related to the change in sensitivity and specificity, it is necessary to select the cut-off value depending on the clinical situation.

## Figures and Tables

**Figure 1 cancers-14-05961-f001:**
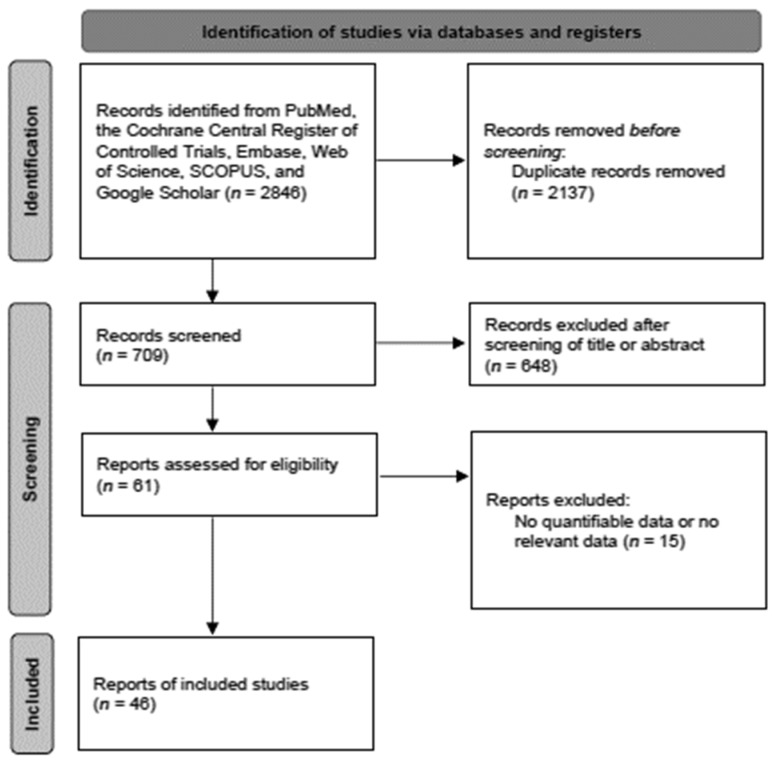
Diagram of selection of studies.

**Table 1 cancers-14-05961-t001:** Diagnostic efficacy and the ROC curves of Kwak TI-RADS categories.

	Sensitivity	Specificity	DOR	AUC
5	0.1408, 95% CI [0.1118; 0.1759]; I^2^ = 93.4%	0.9910, 95% CI [0.9842; 0.9949]; I^2^ = 88.0%	15.4646, 95% CI [8.2029; 29.1549]; I^2^ = 88.3%	0.586
4c	0.7729, 95% CI [0.6786; 0.8458]; I^2^ = 98.2%	0.8277, 95% CI [0.7679; 0.8747]; I^2^ = 98.4%	16.8454, 95% CI [11.1916; 25.3553]; I^2^ = 95.7%	0.871
4b	0.9640, 95% CI [0.9436; 0.9773]; I^2^ = 92.7%	0.5373, 95% CI [0.4511; 0.6213]; I^2^ = 98.8%	31.8051, 95% CI [21.3155; 47.4568]; I^2^ = 88.2%	0.884
4a	0.9918, 95% CI [0.9817; 0.9964]; I^2^ = 95.0%	0.2784, 95% CI [0.2040; 0.3675]; I^2^ = 99.1%	31.1766, 95% CI [19.4820; 49.8914]; I^2^ = 81.2%	0.808

ROC; receiver operating characteristic, TI-RADS; thyroid imaging reporting and data system, CI; confidence index, DOR; diagnostic odds ratio, AUC; area under the receiver operating characteristic curve.

**Table 2 cancers-14-05961-t002:** Diagnostic efficacy and the ROC curves of ACR TI-RADS categories.

	Sensitivity	Specificity	DOR	AUC
TR5	0.7102, 95% CI [0.6156; 0.7895]; I^2^ = 99.0%	0.8696, 95% CI [0.8257; 0.9038]; I^2^ = 98.8%	17.3134, 95% CI [13.3192; 22.5054]; I^2^ = 94.1%	0.875
TR4	0.9437, 95% CI [0.9033; 0.9679]; I^2^ = 97.9%	0.5224, 95% CI [0.4507; 0.5931]; I^2^ = 99.2%	17.5185, 95% CI [12.5652; 24.4244]; I^2^ = 91.7%	0.786
TR3	0.9886, 95% CI [0.9729; 0.9953]; I^2^ = 97.7%	0.2372, 95% CI [0.1680; 0.3239]; I^2^ = 99.5%	15.2758, 95% CI [9.9838; 23.3729]; I^2^ = 86.6%	0.765

ROC; receiver operating characteristic, TI-RADS; thyroid imaging reporting and data system, CI; confidence index, DOR; diagnostic odds ratio, AUC; area under the receiver operating characteristic curve.

**Table 3 cancers-14-05961-t003:** Direct comparison of Kwak (4b) and ACR TI-RADS (TR4) categories.

	Sensitivity	Specificity	DOR	AUC
Kwak TI-RAD (4b)	0.9727, 95% CI [0.9516; 0.9848]; I^2^ = 94.0%	0.5641, 95% CI [0.5009; 0.6253]; I^2^ = 97.1%	44.0619, 95% CI [26.1278; 74.3060]; I^2^ = 91.2%	0.842
ACR TI-RAD (TR4)	0.9760, 95% CI [0.9587; 0.9861]; I^2^ = 92.3%	0.5025, 95% CI [0.4350; 0.5700]; I^2^ = 97.5%	38.2503, 95% CI [24.2254; 60.3949]; I^2^ = 87.0%	0.846
*p* value	0.7572	0.1916	0.6896	0.913

TI-RADS; thyroid imaging reporting and data system, DOR; diagnostic odds ratio, CI; confidence index, AUC; area under the receiver operating characteristic curve.

**Table 4 cancers-14-05961-t004:** Direct comparison of Kwak (4c) and ACR TI-RADS (TR5) categories.

	Sensitivity	Specificity	DOR	AUC
Kwak TI-RADS (4c)	0.7517, 95% CI [0.6875; 0.8064]; I^2^ = 97.0%	0.8754, 95% CI [0.8408; 0.9034]; I^2^ = 96.2%	20.9770, 95% CI [14.4931; 30.3617]; I^2^ = 95.1%	0.89
ACR TI-RADS (TR5)	0.7824, 95% CI [0.6455; 0.8766]; I^2^ = 99.0%	0.8544, 95% CI [0.8096; 0.8901]; I^2^ = 97.1%	21.3275, 95% CI [12.6104; 36.0705]; I^2^ = 97.0%	0.892
*p* value	0.6537	0.4122	0.9597	0.962

TI-RADS; thyroid imaging reporting and data system, DOR; diagnostic odds ratio, CI; confidence index, AUC; area under the receiver operating characteristic curve.

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
