# Peer review of "Diagnostic Performance of ACR and Kwak TI-RADS for Benign and Malignant Thyroid Nodules: An Update Systematic Review and Meta-Analysis"

_cancers, 2022, doi:10.3390/cancers14235961_

Round 1

Reviewer 1 Report

The paper is interesting for supporting the diagnostic accuracy for thyroid malignancies. TIRADs system unfortunately could be influenced by the operator. I think the paper could be useful for the clinician in the clinical practice

Author Response

Dear Reviewers

  We thank again the editor and the reviewers for their comments and appreciate the insightful suggestions to improve the quality of our Manuscript ID # cancers-2018629, ‘Diagnostic Performance of ACR and Kwak TI-RADS for benign and malignant thyroid nodules: A meta-analysis’. We appreciate this opportunity to respond to their concerns and comments.

  We have corrected the manuscript on the basis of the reviewers’ comments and itemize our replies/corrections according to the issues raised by reviewers. We hope that you will find the revised submission to merit its publication in Cancers. Thank you for your consideration. I look forward to hearing from you.

Yours sincerely,

Se Hwan Hwang, M.D., Ph.D.

Department of Otolaryngology-Head and Neck Surgery, Bucheon St. Mary’s Hospital, College of Medicine, The Catholic University of Korea.

Address: 327 Sosa-ro, Bucheon-si, Seoul, 14647, Korea.

Phone: +82 32 340 7044; Fax: +82 32 340 2674; E-mail address: yellobird@catholic.ac.kr

Reviewer 2 Report

The authors compare Kwak-TIRADS and ACR TIRADS to evaluate the diagnostic efficiency of these criteria through a meta-analysis.

The manuscript is well-written and the methods and results are described adequately.

In line 60, the authors mention the following: "In this study, we suggest a more effective diagnostic tool...." This statement is a bit misleading. It gives the sense that the authors have come up with novel diagnostic criteria when the authors mean to say that they have identified the scores with best ROCs in Kwak and ACR TIRADS classifications. I would suggest that the authors rephrase this paragraph along the lines of, "In this study, we compared the diagnostic efficacy of Kwak TIRADS and ACR TIRADS...."

Author Response

(The authors gave the same response as above.)

Reviewer 3 Report

The authors performed a meta-analysis of published studies comparing ACR-TIRADS and KWAK-TIRADS in terms of their performance and clinical utility. The analysis was conducted carefully, taking into account existing requirements. 

Author Response

(The authors gave the same response as above.)
